



# Decomposition of skill scores for conditional verification – Impact of AMO phases on the predictability of decadal temperature forecasts

Andy Richling[1], Jens Grieger[1], and Henning W. Rust[1]

[1]Freie Universität Berlin, Institute of Meteorology, Carl-Heinrich-Becker Weg 6–10, 12165 Berlin, Germany

**Correspondence:** Andy Richling (andy.richling@fu-berlin.de)

**Abstract.** We present a decomposition of skill scores for the conditional verification of weather and climate forecast systems. Aim is to evaluate the performance of such a system individually for predefined subsets with respect to the overall performance. The overall skill score is decomposed into: (1) the *subset skill score* assessing the performance of a forecast system compared to a reference system for a particular subset; (2) the *frequency weighting* accounting for varying subset size; (3) the *reference weighting* relating the performance of the reference system in the individual subsets to the performance of the full data set. The decomposition and its interpretation is exemplified using a synthetic data set. Subsequently we use it for a practical example from the field of decadal climate prediction: An evaluation of the Atlantic-European near-surface temperature forecast from the German initiative Mittelfristige Klimaprognosen (MiKlip) decadal prediction system conditional on different Atlantic Meridional Oscillation (AMO) phases during initialization. With respect to the chosen Western European North Atlantic sector, the decadal prediction system *preop-dcpp-HR* performs better than the un-initialized simulations mostly due to performance gain during a positive AMO phase. Compared to the predecessor system (*preop-LR*), no overall performance benefits are made in this region, but positive contributions are achieved for initialization in neutral AMO phases. Additionally, the decomposition reveals a strong imbalance among the subsets (defined by AMO phases) in terms of *reference weighting* allowing for sophisticated interpretation and conclusions. This skill score decomposition framework for conditional verification is a valuable tool to analyze the effect of physical processes on forecast performance and consequently supports model development and improvement of operational forecasts.

## 1 Introduction

The verification of forecast systems plays an important role in the field of weather and climate prediction to asses the quality of such systems and, moreover, of the entire forecast process. Furthermore, a common practice for evaluating forecast systems is the comparison against a standard reference forecast, e.g., the persistence or climatological forecast, or another competing prediction system. Basically, the relative performance in terms of accuracy of a prediction system with respect to a reference is expressed as forecast skill and is usually presented as a skill score (Wilks, 2011). Therefore, a variety of skill scores are widely used for verification, e.g., the mean squared error skill ccore (MSESS) is a common way to verify a deterministic forecast, while the Brier skill score (BSS), the ranked probability skill score (RPSS) or the continuous ranked probability skill score




(CRPSS), e.g., in decadal forecast verification (e.g., Kadow et al., 2016; Kruschke et al., 2016; Pasternack et al., 2018, 2021), could be the choice for a probabilistic forecast.

Since the forecast performance is typically not homogeneous in time and space, it is of interest how variable the forecast skill is for different states of the system. Therefore, conditional verification is a common practice in weather and climate research, i.e. the evaluation of forecasts separately for different regions (e.g., Northern Hemisphere and Southern Hemisphere) or seasons (e.g., winter and summer). Additionally, the initial state and particular conditions the system goes through during the forecast might also affect the prediction skill. In weather forecasting, the state of atmospheric flow regimes or circulation patterns can influence the forecast quality (Grönås, 1982, 1985), where a more stable regime such as blocking can improve the forecast quality of a model (Tibaldi and Molteni, 1990). The presence of different climate states during the initialization procedure of medium-range forecasts, which can improve the predictive ability in certain periods, is addressed in the subseasonal-to-seasonal (S2S) prediction community (Mariotti et al., 2020). Large-scale atmospheric circulation variability, such as the North Atlantic Oscillation (NAO, Jones et al., 2004; Ferranti et al., 2015; Jones et al., 2015), the Madden-Julien Oscillation (MJO, Ferranti et al., 2018) or circulation patterns (Frame et al., 2013; Richardson et al., 2021) as well as coupled ocean-atmospheric phenomena like El Niño–Southern Oscillation (ENSO, e.g., Qin and Robinson, 1995; Branković and Palmer, 2000; Goddard and Dilley, 2005; Frías et al., 2010; Kim et al., 2012; Manzanas et al., 2014; Miller and Wang, 2019) can contribute to a forecast skill improvement. In decadal climate prediction – the focus of this study – the state of the ocean has the potential to affect long-term forecasts of the following years, i.e. an enhanced subpolar ocean heat transport (OHT) linked to North Atlantic upper ocean heat content (UOHC) and in some way via the Atlantic Meridional Overturning Circulation (AMOC) to the positive Atlantic Multidecadal Oscillation/Variability (AMO/AMV) phase shows the potential of an improved predictive ability during the initialization of a climate model (Müller et al., 2014; Zhang and Zhang, 2015; Borchert et al., 2018, 2019).

In a typical verification study, the accuracy of a given forecast is compared to a reference to evaluate the quality of the forecast. To assess the forecast quality for specific situations (states, seasons, regions, etc.) verification can be carried out conditional on these situations by stratifying the full data set along situations. Thus the forecast data set is split up and (skill) scores are obtained individually for the splits. The interpretation of these partial skill scores is not necessarily straightforward. This is particularly the case when the reference strongly varies among individual subsets compared to the overall behavior and is commonly known as "Simpson's Paradox" (Pearson et al., 1899; Yule, 1903; Simpson, 1951; Blyth, 1972). With respect to weather and climate prediction, a potential mis-interpretation of the forecast performance stratified along specific conditions or samples may arise if the underlying climatology that is used as reference forecast differs in some way among these samples (e.g., Murphy, 1996; Goeber et al., 2004; Hamill and Juras, 2006). In that case a fair comparison should consider the varying behavior of such climatology in the verification procedure.

While the majority of mentioned studies focus more on decomposing a skill score to measure basic aspects of forecast quality with respect to a climatological reference forecast in a fair way, here we apply a decomposition framework in the context of conditional verification in the field of decadal predictions. The aim is to evaluate the performance of individual subsets in relation to the performance of the entire forecast set. The decomposition provides a simple diagnostic tool to assess the contribution of certain subsets to the overall skill as well as to identify potential causes of variable skill between these





subsets. The resulting information can be further used to analyze physical processes related to certain subsets and consequently to support the model development and to optimize operational forecasts. In terms of decadal forecasts, we exploit the potential source of long-term predictability forced by ocean states associated with the AMO to improve the forecast assessment.

First, the general decomposition procedure of the skill score is described in section 2 and exemplified in section 3 using synthetic data. In section 4, the decomposition is applied to decadal predictions to evaluate the Atlantic-European near-surface temperature forecast of a pre-operational forecast system depending on different North Atlantic ocean states. The latter are determined by the Atlantic Meridional Oscillation (AMO). The results are summarized and discussed in section 5. Section 6 concludes this study.

## 2  Decomposition of skill score

This section presents the decomposition of a skill score into contributions from different subsets derived from the full set of forecast-observation pairs and discusses the interpretation of individual terms.

### 2.1  Subset contribution

To verify a forecast $f_n$ we calculate a verification score $\mathrm{S}_n(f_n, o_n)$, an error metric between an individual forecast $f_n$ and

the corresponding observation $o_n$ (Wilks, 2011). Considering all forecast-observation pairs $(f_n, o_n), n = \{1, \ldots, N\}$, the mean score $\mathrm{S}$ of the full set can be computed by

$$\mathrm{S} = \frac{1}{N} \sum_{n=1}^{N} \mathrm{S}_n(f_n, o_n). \tag{1}$$

The mean squared error (MSE) is an adequate score for a deterministic forecast of a continuous variable, while the ranked probability score (RPS) is an appropriate choice for a probabilistic forecast of a discrete forecast. To measure the performance

of a forecast system $fc$ compared to a reference forecast $ref$, the associated skill score SS (e.g., MSE skill score MSESS and ranked probability skill score RPSS, respectively) is used.

The forecast performance may vary for individual subsets of the data and the resulting interpretation may depend on the different behavior of the reference system. To assess varying skill scores for specific situations (e.g., states, time periods, seasons, regions, etc.), the verification is carried out conditional on these situations, i.e., the full data set is stratified. We thus





split the data into $K$ subsets and determine the individual contribution of each subset $i$ to the overall mean skill score SS

$$SS = \frac{S^{fc} - S^{ref}}{S^{perf} - S^{ref}}$$

$$= \frac{\sum_{i=1}^{K} \frac{N_i}{N} S_i^{fc} - \sum_{i=1}^{K} \frac{N_i}{N} S_i^{ref}}{S^{perf} - S^{ref}} \tag{2}$$

$$= \sum_{i=1}^{K} \frac{N_i}{N} \underbrace{\left( \frac{S_i^{fc} - S_i^{ref}}{S^{perf} - S^{ref}} \right)}_{\text{contribution subset i}},$$

where $S^{fc}$ and $S^{ref}$ is the mean score of the forecast system *fc* and the reference system *ref*, respectively, over an entire data set with $N$ forecast-observation pairs and $S^{perf}$ the score of a perfect forecast, which is 0 for the MSE or RPS. $S_i^{fc}$ and $S_i^{ref}$ represent the mean score of the forecast system and reference system, respectively, for individual subsets, where $N_i$ is the number of
forecast-observation pairs in subset $i$.

## 2.2 Terms of decomposition

In order to evaluate how strongly and in which situations the skill score of the subsets affects the total skill score, we include and separate any component that influences the contribution of a subset to the overall skill score. We multiply equation Eq. (2) by $1 = \frac{S^{perf} - S_i^{ref}}{S^{perf} - S_i^{ref}}$, yielding

$$SS = \sum_{i=1}^{K} \underbrace{\frac{N_i}{N}}_{\substack{\text{frequency} \\ \text{weighting}}} \cdot \underbrace{\left( \frac{S_i^{fc} - S_i^{ref}}{S^{perf} - S_i^{ref}} \right)}_{\substack{\text{subset} \\ \text{skill score}}} \cdot \underbrace{\left( \frac{S^{perf} - S_i^{ref}}{S^{perf} - S^{ref}} \right)}_{\substack{\text{reference} \\ \text{weighting}}}$$

$$= \sum_{i=1}^{K} W_{\text{freq}_i} \cdot SS_i \cdot W_{\text{ref}_i} = \sum_{i=1}^{K} W_i \cdot SS_i . \tag{3}$$

This decomposition of the total skill score results in three terms characterizing the contribution of a subset to the overall skill score:

### 2.2.1 Subset skill score

$SS_i = \frac{S_i^{fc} - S_i^{ref}}{S^{perf} - S_i^{ref}}$ gives the mean *subset skill score* of the forecast system *fc* versus the reference system *ref* with respect to
forecast-observation pairs of the given subset $i$. This term characterizes how well the forecast system performs in comparison to the reference system *in that specific subset*, e.g., during a positive AMO phase. It is commonly applied in model evaluations to find enhanced predictability during certain climate or large-scale circulation states or specific seasons. In Sect. 3, this term can be found as $SS_{1/2}$ in Table 1 and 2 as well as in Fig. 1.



### 2.2.2 Frequency weighting

$W_{\text{freq}_i} = \frac{N_i}{N}$ considers the number of forecast-observation pairs (e.g., time steps) in subset $i$ relative to the total number of forecast-observation pairs. For a time series one could imagine, this part reflects the relative frequency of occurrence of the situation stratified along within the total time period and is therefore named as *frequency weighting*. Consequently, a situation which does not occur very often will contribute less to the overall skill score compared to an event which occurs more frequently.

### 2.2.3 Reference weighting

$W_{\text{ref}_i} = \frac{\text{S}^{\text{perf}} - \text{S}_i^{\text{ref}}}{\text{S}^{\text{perf}} - \text{S}^{\text{ref}}}$ is the ratio of the mean score of the reference system for the subset $i$ (numerator) and the full set of forecast-observation pairs (denominator). It ajust the scale (or range) of the subset skill score which was set by $\text{S}^{\text{perf}} - \text{S}_i^{\text{ref}}$ to the scale used for the overall skill score. This component can be interpreted as a weighting of the subset skill score by means of the performance of the reference system in the subset compared to its performance in the full set of forecast-observation pairs. If

the performance of the reference varies strongly among subsets, the individual subset skill scores will contribute to the total skill score according to the performance of the reference. We call this component *reference weighting*.

### 2.2.4 Subset weighting

In summary, the individual contribution of a certain subset to the overall skill score depends on i) the performance of the forecasting system compared to the reference system in that given subset, weighted by ii) the relative size of the subset (frequency

of the stratification event occurring) and iii) the performance of the reference system in the subset compared to the full set of forecast-observation pairs. The total subset weight

$$W_i = \frac{N_i}{N} \cdot \frac{\text{S}^{\text{perf}} - \text{S}_i^{\text{ref}}}{\text{S}^{\text{perf}} - \text{S}^{\text{ref}}} \tag{4}$$

is the product of the *frequency weighting* and the *reference weighting* and determines the influence of the subset on the total skill score, i.e. for an improvement/degeneration $\Delta\,\text{SS}_i$ of the forecast in the subset $i$, the total skill score for the full set of

forecast-observation pairs changes by

$$\Delta\,\text{SS}_{(\Delta\,\text{SS}_i)} = \frac{N_i}{N} \cdot \frac{\text{S}^{\text{perf}} - \text{S}_i^{\text{ref}}}{\text{S}^{\text{perf}} - \text{S}^{\text{ref}}} \cdot \Delta\,\text{SS}_i \;. \tag{5}$$

## 3 Synthetic time series cases

We illustrate the effect of the different reference performance using a synthetic data set in the following. In the context of near-term climate prediction one could imagine the annual mean of 2m-temperature being verified in two different forecast

systems with respect to the same observation for a certain defined time period.





| Case | (Skill) score behavior | $S_1^{fc}$ | $S_1^{ref}$ | $SS_1$ | $S_2^{fc}$ | $S_2^{ref}$ | $SS_2$ | $S^{fc}$ | $S^{ref}$ | $SS$ |
|------|------------------------|-----------|-------------|--------|-----------|-------------|--------|----------|-----------|------|
| *A0* | $SS_1$ is worse compared to $SS_2$; SS close to $SS_2$ | 0.30 | 0.22 | **-0.36** | 1.56 | 2.69 | **0.42** | 0.93 | 1.46 | **0.36** |
| *A1* | $SS_1$: increase; $SS_2$: unchanged; SS: nearly unchanged | 0.18 | 0.22 | **0.18** | 1.56 | 2.69 | **0.42** | 0.87 | 1.46 | **0.40** |
| *A2* | $SS_1$: increase; $SS_2$: decrease; SS: decrease | 0.13 | 0.22 | **0.41** | 3.46 | 2.69 | **-0.29** | 1.93 | 1.46 | **-0.23** |

**Table 1.** Different cases (*A0-A2*) showing the scores (S) and skill scores (SS) of two subsets and of the total forecast time series. The contribution of subset 1 to the total skill score is weak compared to the contribution of subset 2. The (skill) score changes as described in *A1* and *A2*, both related to the first case *A0*.

## 3.1 Example cases with different behavior of skill scores

With respect to a time-based stratified verification which is addressed in this study, we assume that the performance of both forecast systems varies systematically within the time period considered. For this purpose, we divide the entire time period – here we use a time period of $N = 60$ time steps representing 60 years – into two equal sized subsets ($K = 2$, $N_1 = N_2 = 30$). The performance of the two forecast systems shows a systematically different behavior for the two subsets. An example from near-term climate prediction could be the state of the ocean in terms of years dominated by a negative or positive AMO phase during the initialization procedure, which might have an influence on the forecast performance in some regions via the OHT (Borchert et al., 2018).

Applied to our fictive example, the mean score of the forecast systems differs between both subsets ($S_1^{fc} \neq S_2^{fc}$). The same assumption holds for the mean score of the reference system ($S_1^{ref} \neq S_2^{ref}$). In some situations it is possible that the long-term performance expressed in terms of total skill score SS of a forecast system compared to another forecast system is dominated by a specific subset period. With the setup described above and the decomposition approach from Sect. 2, we illustrate and discuss the individual contributions of subsets to the total skill score. For this purpose we generate six hypothetical cases with different performance combinations of forecast *fc* and reference *ref* during the two subsets $i = 1$ and $i = 2$. Case *A* assumes a very different performance of the reference system in the two subsets and case *B* assumes an almost equal performance of the reference instead.

### 3.1.1 Case A: Unequal performance of the reference

In the first case (*A0*, see Table 1) we assume the forecast system *fc* performs poorly compared to the reference in subset $i = 1$ (subset skill score $SS_1 = -0.36$). In contrast, forecast system *fc* performs better compared to the reference in subset $i = 2$ (subset skill score $SS_2 = 0.42$). As a first guess from seeing the skill scores one might assume the total skill score SS being an equal composition (e.g., arithmetic mean) of both subset skill scores $SS_{1/2}$ leading to a value close to zero. However, in this specific configuration the total skill score of the overall data (SS = 0.36) is very close to that one in subset 2. The total skill of forecast system *fc* is mainly dominated by this subset.




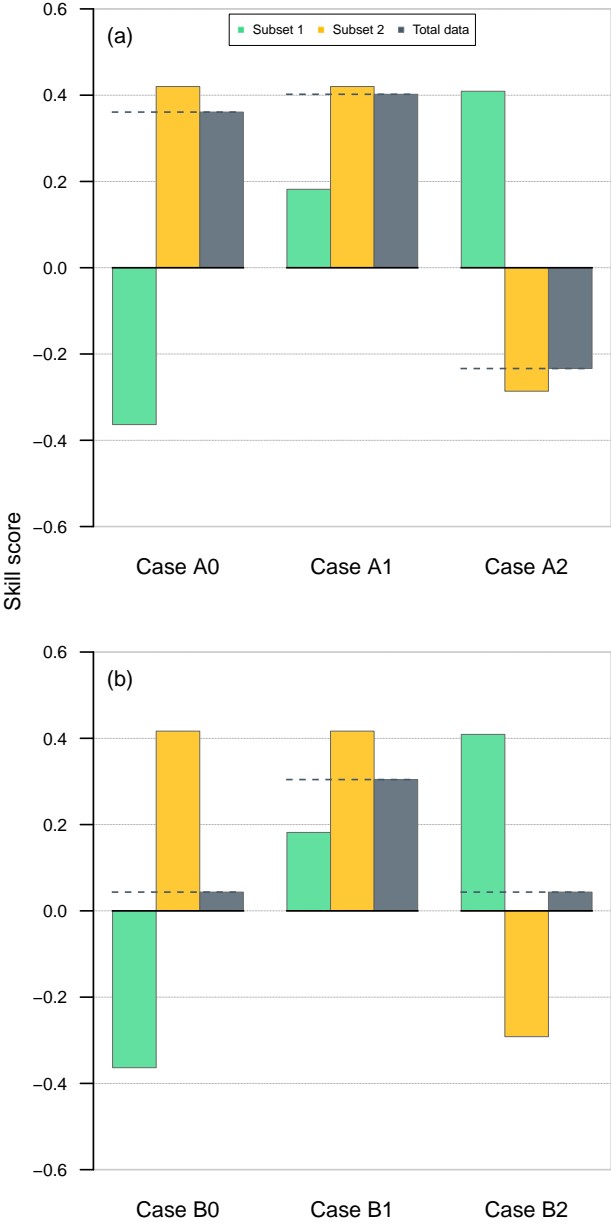

**Figure 1.** Subset skill scores (green/orange bars) and their influence on the respective total skill score (gray bars and dashed lines) from synthetic example cases of (a) setup *A* shown in Table 1 (strong reference weighting imbalance among both subsets) and (b) setup *B* shown in Table 2 (nearly balanced reference weighting among both subsets).





| Case | (Skill) score behavior | $S_1^{fc}$ | $S_1^{ref}$ | $SS_1$ | $S_2^{fc}$ | $S_2^{ref}$ | $SS_2$ | $S^{fc}$ | $S^{ref}$ | $SS$ |
|---|---|---|---|---|---|---|---|---|---|---|
| *B0* | $SS_1$ is worse compared to $SS_2$; SS close to zero | 0.30 | 0.22 | **-0.36** | 0.14 | 0.24 | **0.42** | 0.22 | 0.23 | **0.04** |
| *B1* | $SS_1$: increase; $SS_2$: unchanged; SS: increase | 0.18 | 0.22 | **0.18** | 0.14 | 0.24 | **0.42** | 0.16 | 0.23 | **0.30** |
| *B2* | $SS_1$: increase; $SS_2$: decrease; SS: unchanged | 0.13 | 0.22 | **0.41** | 0.31 | 0.24 | **-0.29** | 0.24 | 0.23 | **0.04** |

**Table 2.** Cases *B0-B2* similar to table 1, but in contrast the contribution of subset 1 and subset 2 to the total skill score is similar.

From just focusing on subset skill scores, one could be tempted to improve the forecast system *fc* especially for subset 1
where the skill score compared to the reference system is worse. This scenario will be covered in case *A1* (Table 1) where an improvement of the subset skill score is achieved for the first subset, while the skill score of the second subset remains the same. Although the skill score of the forecast system *fc* in subset $i = 1$ is improved (*A1*: $SS_1 = 0.18$), the overall skill score hardly changes (*A1*: SS = 0.40).

In the last case (*A2* in Table 1), we simulate a stronger improvement of the skill score in subset $i = 1$ compared to case *A1*
($SS_1 = 0.41$), which is accompanied by a reduction in skill score for subset $i = 2$ ($SS_2 = -0.29$). Here, the total skill score (SS = $-0.23$) decreases compared to *A0* although there is a stronger improvement in subset $i = 1$ than the decline in the second subset.

Taking into account all three cases, it can be summarized that the total skill score of the forecast system *fc* with respect to the reference *ref* is mainly dominated by the subset skill score from subset $i = 2$; to be seen in Fig. 1a, where the overall skill
score of the full set of forecast-observation pairs (gray bars) behaves very sensitively towards changes in the subset skill score from subset $i = 2$ (orange), whereas changes of the skill score from subset $i = 1$ (green) yield almost no effect.

### 3.1.2 Case B: Equal performance of the reference

In contrast to case A we show three additional examples (*B0-B2* in Table 2) in which the influence on the total skill score is nearly equally balanced between both subsets. To see the different behavior, the subset skill scores of all cases will be the same
as before. Consequently, in the first case (*B0* in Table 2) the forecast system *fc* performs worse in subset $i = 1$ compared to the reference system *ref* ($SS_1 = -0.36$), while it shows a better performance in subset $i = 2$ ($SS_2 = 0.42$). Unlike case *A0*, the total skill score now depends almost equally on both subsets (SS = 0.04). The changes made to the two cases *B1* and *B2* follow a similar pattern as the changes in *A1* and *A2* as can be seen in Fig. 1b, whereas the total skill score is almost given by the arithmetic mean of both periods.

With the skill score decomposition from Sect. 2 the reason for this behavior can be investigated.

### 3.2 Decomposition of skill scores and impact of the reference weighting

The different behaviors shown can be investigated using the decomposition terms from Eq. (3) with $S^{perf} = 0$. As demonstrated there, the contribution of an individual subset to the total skill score depends on three terms: frequency weighting, reference



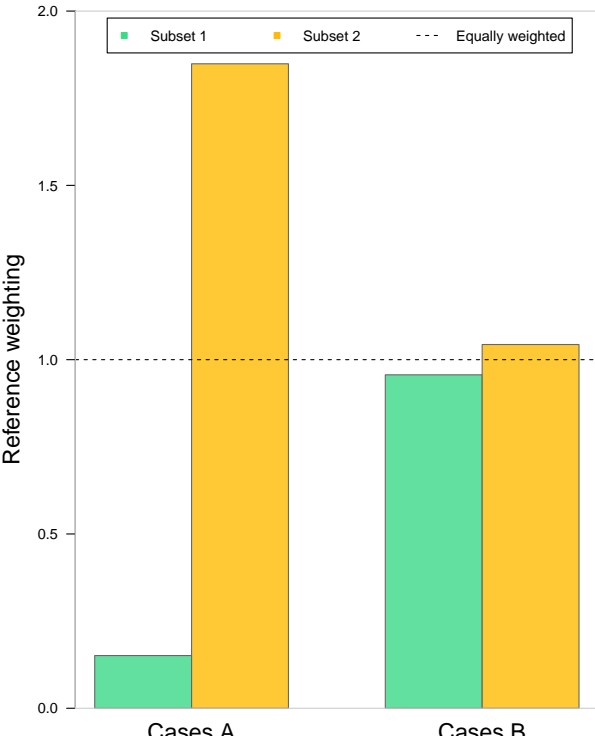

**Figure 2.** Reference weighting of both subsets (green/orange bars) for synthetic case setup *A* and *B*. Cases from setup *A* show strong reference weighting imbalance among both subsets and cases of setup *B* show nearly balanced reference weighting among both subsets. The dashed line reflects a balanced behavior among both subsets.

weighting and the subset skill score. As defined above, we varied the subset skill scores in a same way and used equal-sized subsets resulting in the same frequency weighting of $\frac{1}{2}$ for both subsets. Consequently, the reference weighting for the individual subsets must play a crucial role. For case *A* the scores ($S_{1/2}$) between subsets differ by more than one unit. In detail, the scores generally are much higher in subset $i = 2$ than in subset $i = 1$. As a result, potential subset skill score changes for the forecast system *fc* that are just achieved during the first subset will not affect the total skill score very much. The larger scores in subset $i = 2$ show a stronger relevance with respect to the total skill.

In contrast to setup *A*, the cases generated in *B* show a nearly balanced behavior in this respect. These difference can also be seen when we compare the reference weighting term from the skill score decomposition described before. Figure 2 visualizes this behavior, in which the cases from setup *A* show a different value for the reference weighting in both subsets, while in setup *B* the reference weighting is close to 1 in both cases.

Gnerally, the reference weighting lies between 0 and $K$ (number of subsets). Values below 1 reflect a lower than average contribution to the overall skill score while values above 1 indicate a higher than average contribution. Figure 3 demonstrates





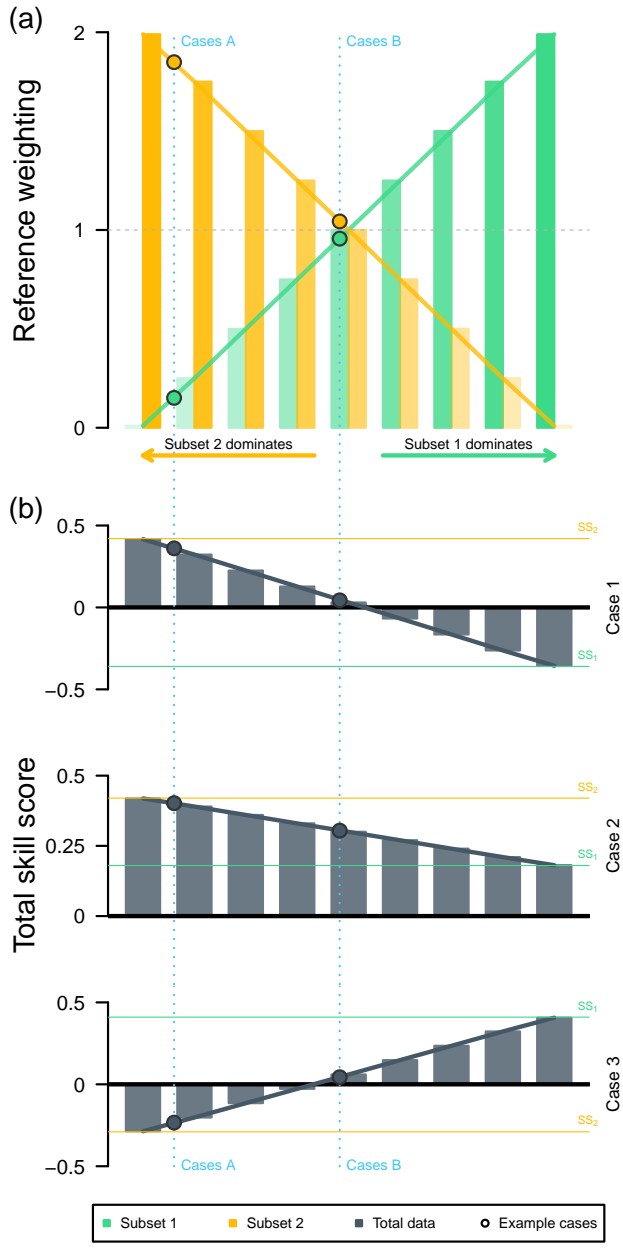

**Figure 3.** (a) Variations in the reference weighting term of both subsets (green/orange bars) and (b) their potential influence on the corresponding total skill score (gray bars) for given subset skill scores $SS_{1/2}$ (green/orange horizontal lines) from example cases *A0-A2* described in Table 1 and *B0-B2* in Table 2. Current values of the example cases are highlighted with a dot. A balanced (enhanced unbalanced) behavior among both subsets reflects the center bar pair (bar pairs towards left/rights edges).





| $\Delta\mathrm{SS}_1$ | $\Delta\mathrm{SS}_2$ | $\Delta\mathbf{SS}$ | $W_{\mathrm{ref},1}$ | $W_{\mathrm{ref},2}$ | $W_{\mathrm{freq},1/2}$ |
|---|---|---|---|---|---|
| 0.5 | 0 | **0.04** | 0.15 | 1.84 | 0.5 |
| 0 | 0.5 | **0.46** | 0.15 | 1.84 | 0.5 |

**Table 3.** Individual effect of a 0.5 change in the subset skill score $\mathrm{SS}_i$ on the total skill score SS for example case *A0*. The weighting terms from the decomposition are also shown.

the impact of individual subset skill scores on the resulting total skill score depending on their reference weighting. For this purpose, we compute the total skill score SS with respect to our cases (Fig. 3b) with prescribed subset skill score in subset $i = 1$ ($\mathrm{SS}_1$) and subset $i = 2$ ($\mathrm{SS}_1$), respectively, and change successively the reference weighting term (Fig. 3a). Starting with a behavior similar to the setup *A* which is dominated by subset $i = 2$ (left in Fig. 3), where the reference weighting term of subset $i = 2$ (orange bars) is larger than the one of subset $i = 1$ (green). A balanced ratio between both subsets (similar to case *B*) is shown in the middle; The right part shows a total skill score which is mainly controlled by the subset $i = 1$. Thus, the ratio of the score of the reference system between a subset and the full data set – captured here by the reference weighting term – controls the subset's contribution to the overall skill score.

According to Eq. (5), we can compute potential changes of the total skill score $\Delta\,\mathrm{SS}$ depending on changes in the subset skill score $\Delta\,\mathrm{SS}_i$. For example, in case *A0* a change of the subset skill score in subset $i = 1$ of $\Delta\,\mathrm{SS}_1 = 0.5$ (e.g., increase of $\mathrm{SS}_1$ from -0.36 to 0.14) would change the total skill score of only $\Delta\,\mathrm{SS} = 0.04$. On the other hand, a skill gain of 0.5 in subset $i = 2$ would increase the total skill score by a value of 0.46. In detail, with $\mathrm{S}^{\mathrm{perf}} = 0$ the derived weighting terms from the decomposition are shown in Table 3. In this example, it is more effective in terms of gain in total skill score to focus on the subset $i = 2$ for improvement of the forecast system.

The synthetic example is focused on the reference weighting; however the decomposition is also useful for unequal subset sizes. The contribution to the total skill score is then additionally controlled by the frequency weighting. Depending on the verification setup, both parts should be considered in weather and climate forecasts. As a consequence, complexity is reduced when each subset has the same size and the reference weighting of all subsets is 1 due to a chosen reference. This leads to equally weighted skill scores of the subsets.

## 3.3 Subset contributions

In Fig. 4, we assess the subset contributions compared to a balanced contribution across the synthetic example cases. The balanced contribution (gray horizontal lines) represents a hypothetical value resulting from distributing the total skill score into equal contributions from the $K$ subsets: $\mathrm{SS}_{\mathrm{bal}} = \frac{\mathrm{SS}}{K}$. The sign of the subset contribution indicates positive or negative contribution to the total skill score, while its value indicates the amount of the contribution. In setup *A*, the absolute values of the contribution from subset 2 (orange bars) are much larger, while the contributions of subset 1 (green bars) remain negligibly small. The large deviations from $\mathrm{SS}_{\mathrm{bal}}$ shows the strong imbalance between the contributions of both subsets. Since, the



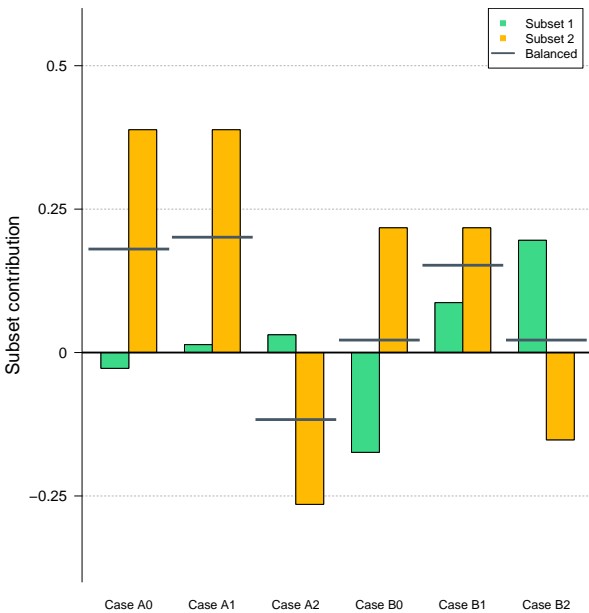

**Figure 4.** Subset contributions of both subsets (green/orange bars) from cases *A0-A2* and *B0-B2*. The sign of the bars accounts for positive or negative contributions to the total skill score. Gray horizontal lines indicate a balanced contribution with respect to the total skill score.

frequency weighting of both subsets is identical, the observed characteristic is driven by the reference weightings. In contrast, the strengths of the subsets' contributions in setup *B* are more similar, hence the subsets' skill scores directly affects the subsets' contributions without relevant modifications from the weighting terms.

In summary, the decomposition of the subset contribution into its three componentes reveals the potential impact of a subset on the overall skill score considering the combination of all three terms of the subset (i.e., size and performance of the reference) instead of only the skill score for a particular subset.

## 4    Conditional verification in the MiKlip decadal prediction system

### 4.1    Simulations from MiKlip decadal prediction system

We investigate the influence of ocean states – given in terms of AMO phases – on the near-surface air temperature hindcast skill in the MiKlip decadal climate prediction system. The MiKlip decadal climate prediction system (Marotzke et al., 2016) generation *preop-dcpp* is based on the coupled atmosphere-ocean Earth System Model of the Max-Planck Institute version 1.2 simulated in the high-resolution setting *HR* (Müller et al., 2018; Mauritsen et al., 2019). The model for the atmospheric component – ECHAM6.3 – has a T127 horizontal resolution ($0.9375°$) and 95 vertical levels. The ocean part is simulated by the Max Planck Institute ocean model (MPIOM) with a horizontal resolution of $0.4°$ and 40 vertical levels.



The 10-member ensemble of the system is initialized on an annual basis from 1960 to 2012, with a period of 10 years being simulated for each run. The initialization procedure is similar to Pohlmann et al. (2013) which is based on nudging the model toward atmospheric and oceanic fields obtained from reanalysis data. In case of the atmospheric model component, a full atmospheric field-initialization from ERA40 (Uppala et al., 2005) and Era-Interim (Dee et al., 2011) reanalyses is applied. For the

ocean component, salinity and ocean temperature anomalies derived from an assimilation experiment forced by ORA-S4 ocean reanalysis data (Balmaseda et al., 2013) as well as sea ice concentrations from the National Snow and Ice Data Center (Fetterer et al., 2018) described in Bunzel et al. (2016) are taken as initial conditions. The external forcing is based on the CMIP6 forcing (see Eyring et al. (2016) and Pohlmann et al. (2019b) for details). In addition to the initialized simulations, an ensemble of 10 uninitialized runs (historical simulations) is used as the competitive hindcast for the skill assessment. Further details about the

simulations can be found in Müller et al. (2018) and Pohlmann et al. (2019b). To evaluate the probabilistic hindcast skill, both hindcast sets are verified against observations from the Hadley Centre and Climate Research Unit (HadCRUT)4 (Morice et al., 2012) on the basis of monthly mean temperatures. To be on the same horizontal resolution as the observational data, the model data of the prediction system is re-gridded to a regular $5° \times 5°$ grid.

## 4.2 Atlantic Multidecadal Oscillation time series

Since the conditional verification of the temperature will be stratified along three different phases of the Atlantic Multidecadal Oscillation (AMO), we calculate the AMO index proposed by Enfield et al. (2001) in the ORA-S4 ocean reanalysis data to match the current state of the Atlantic ocean during the initialization procedure. Specifically, monthly anomalies (base period: 1960–2010) of the sea surface temperature (SST) averaged over the North-Atlantic region (80–0° W, 0–60° N) are exploited to compute the North-Atlantic temperature time series. Afterwards, the linear trend (base period: 1960–2010) of this time series is

removed to obtain the AMO time series. With regard to the subsequent conditional evaluation of the decadal prediction system, annually averages of the AMO are used to split the entire time period into three different subsets (based on $0 \pm 0.5\sigma$ thresholds using base period 1960-2010) representing years of negative, neutral and positive AMO phases.

## 4.3 Verification of probabilistic forecasts for three categories

We verify the decadal ensemble predictions using the ranked probability score RPS (e.g., Wilks, 2011; Kruschke et al., 2016).

The score is computed for both hindcast sets against the HadCRUT observation to asses the probabilistic skill of the initialized vs un-initialized simulations. For near-surface air temperature, we build time series of forecast-observation pairs depending on lead-time for all initialized decadal experiments from 1960 to 2012. Data with lead-times between 2 and 5 years are averaged to obtain a score for the lead-time period 2–5 years.

In the next step we divide the resulting data sets (separately for initialized, un-initialized and observational data) along

their terciles into three equal parts to obtain $J = 3$ different temperature categories $j = 1, \ldots, J$ (below normal, normal, above normal). With this approach, an implicit lead-time-dependent bias-correction, which is commonly applied in decadal climate predictions projects, will be achieved.



The ranked probability score (RPS) defined as

$$\text{RPS}_t = \sum_{j=1}^{J} (Y_{j,t} - O_{j,t})^2 \tag{6}$$

is calculated between both hindcast sets and the observational data, where $Y_{j,t}$ is the cumulative forecast probability of class $j$ (with $J = 3$) derived from the forecast ensemble of initialization year $t$ for the given forecast lead-time mean 2–5 years. $O_{j,t}$ represents the corresponding observed cumulative probability represented as the Heaviside step function where either $O_{j,t} = 0$ in case a higher category than $j$ is observed or $O_{j,t} = 1$ otherwise. To asses the skill between the initialized ($fc$) and un-initialized ($ref$) simulations, the ranked probability skill score

$$\text{RPSS} = 1 - \frac{\overline{\text{RPS}}_{\text{fc}}}{\overline{\text{RPS}}_{\text{ref}}} \tag{7}$$

is computed.

With respect to the conditional verification using the decomposition of the skill score, here we want to evaluate the probabilistic hindcast skill stratified along three phases (negative, neutral, positive) of the AMO instead of two demonstrated in Sect. 2. That means, the RPS and RPSS of the entire time period contain every initialization year $t$ from 1960 to 2012 as time

step, while the AMO-phase-specific terms only consider initialization years which are related to the associated AMO phase.

The information about the significance of the RPSS is based on a 5-year-block-bootstrap method by a 1000-fold re-sampling of the forecast-reference-observation cases in the entire time period. The RPSS value is considered statistically significant if 0 is outside the 95 % inner values of the bootstrap distribution.

A large part of the routines used for verification presented here is implemented as the verification plug-in *ProblEMS* (https:

//www.xces.dkrz.de/plugins/problems/detail/; via guest login; last access: 24 October 2023) into the MiKlip and ClimXtreme Central Evaluation System (https://www.xces.dkrz.de; last access: 24 October 2023) – based on the Free Evaluation System Framework for Earth System Modeling (Freva; Kadow et al., 2021).

### 4.4 Subset contributions of RPSS

Figure 5a shows the RPSS over the European region for the initialized decadal hindcast with respect to the un-initialized hind-

cast averaged over the entire hindcast period for lead-years 2–5. Significant values (marked with a cross) are rare. Negative significant values can be found in the Barents Sea and a larger patch in the south-western North Atlantic. The latter is presumably caused by a displacement of ocean currents in that area since the region is especially sensitive to initializations (Kröger et al., 2018; Polkova et al., 2019). Positive significant skill can be found in the Greenland Sea. Besides individual grid points with significant values only patches with positive but non-significant skill is visible in the eastern Mediterranean and in the

north-eastern part of the North Atlantic.

To exemplify the stratified verification, Fig. 5b-d shows the subset contributions to the total RPSS during (b) negative, (c) neutral and (d) positive AMO phases following Eq. (2). The AMO neutral phase (Fig. 5c) contributes to negative RPSS in the south-western North Atlantic and Barents Sea, while positive contributions in W-EU and C-EU are found during negative AMO phase (Fig. 5b) as well as in the North Atlantic under positive AMO conditions during the initialization procedure (Fig. 5d).



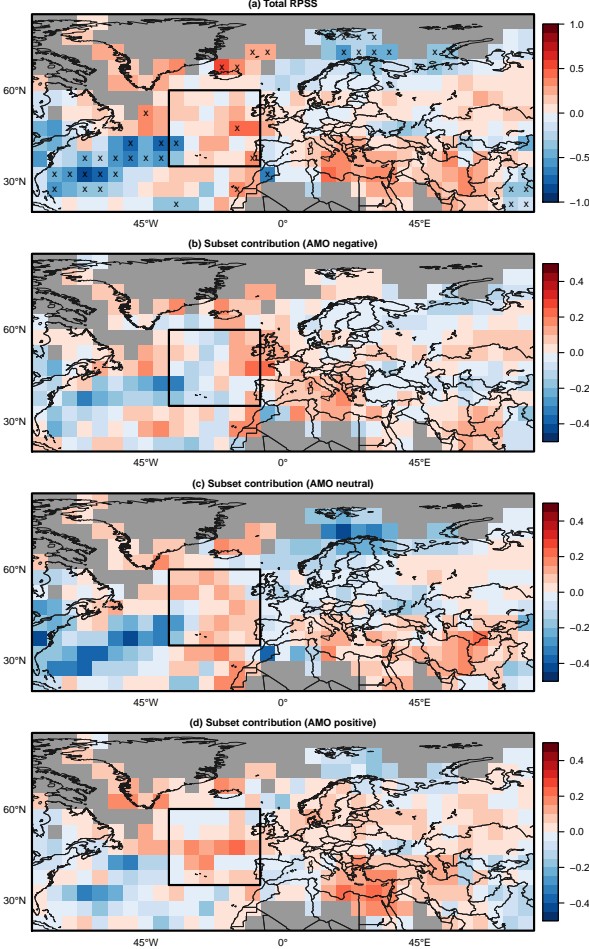

**Figure 5.** (a) Ranked probability skill score (RPSS) of near-surface temperature of the initialized hindcast (*preop-dcpp-HR*) with respect to un-initialized historical simulations and HadCrut observations for lead-year 2–5 from 1962–2017. Additionally, individual contribution of subsets for (b) negative, (c) neutral and (d) positive AMO phase initialization. Missing values are depicted in gray. Crosses in (a) indicate areas with significant (95 %-level) skill scores. The box highlights the W-EU NA region analyzed in Sect. 4.5.

## 4.5 Decomposition of RPSS over Western European North Atlantic

Next, we focus on the Western European North Atlantic region (W-EU NA). This is motivated on the one hand by the different predictability associated with certain AMO phases identified in previous studies, and on the other hand by the positive total skill found in that region. We investigate the subset contributions and the three terms (subset skill score, frequency weighting, reference weighting) of the decomposition for the annual field-mean of the W-EU NA region 40–10° W, 35–60° N, (box in Fig. 5) according to Eq. (2) and (3). The subset skill score (subset RPSS) in Fig. 6b shows no or at most very weak improvement of the initialized prediction system over the un-initialized simulations under negative and neutral AMO conditions during the





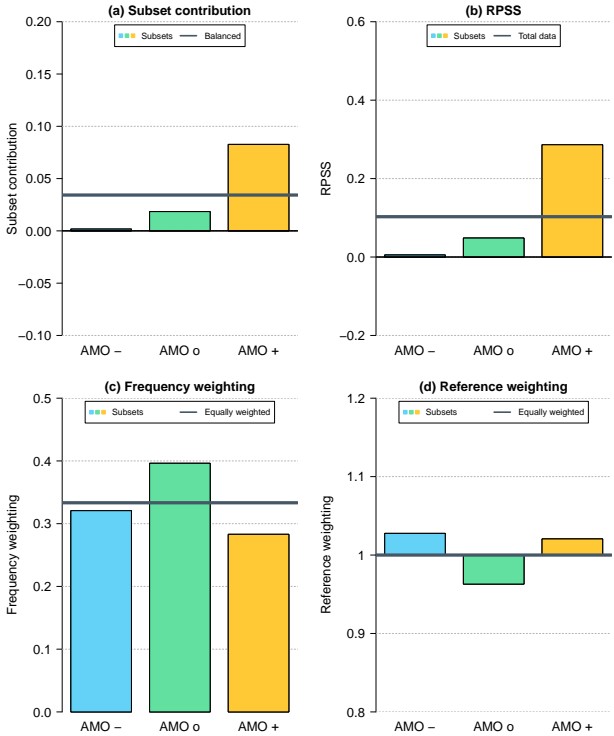

**Figure 6.** (a) Subset contributions related to Eq. (2) as well as (b) subset RPSS, (c) frequency weighting and (d) reference weighting of subsets (defined by AMO initialization) according to Eq. (3) for the conditional verification of near-surface temperature in the W-EU NA region between initialized hindcast (*preop-dcpp-HR*) and un-initialized historical simulations with respect to HadCrut observations for lead-year 2-5 from 1962-2017. Gray horizontal lines represent (b) total skill score, (c–d) balanced weightings and (a) balanced contributions with respect to the total skill score.

initialization procedure. In contrast, a clearly positive subset RPSS of 0.3 is achieved for initialization during positive AMO years. As comparison, the total RPSS is around 0.1 (gray horizontal line). The frequency weighting (Fig. 6c) indicates that initialization years with neutral AMO phase are more frequent (0.4) than years of the other two phases. This leads to a higher

frequency weighting factor associated with the AMO neutral phase. Figure 6d shows that the reference weighting is close to 1 for all phases. As this component represents a potentially different score of the reference system along the three subsets, we do not expect large variability as the uninitialized reference is not be influenced by the AMO phase.

Multiplying the three components for the individual subsets, we arrive at the subset contributions (Fig. 6a). This contribution is mainly determined by the subset skill score (Fig. 6a) and to a small extent modified by the frequency weighting (Fig. 6d).

The resulting subset contributions related to the AMO phases show that the positive AMO phase contributes to a large amount (around 0.08) to the total RPSS, followed by the neutral AMO phase with a smaller contribution of 0.02.



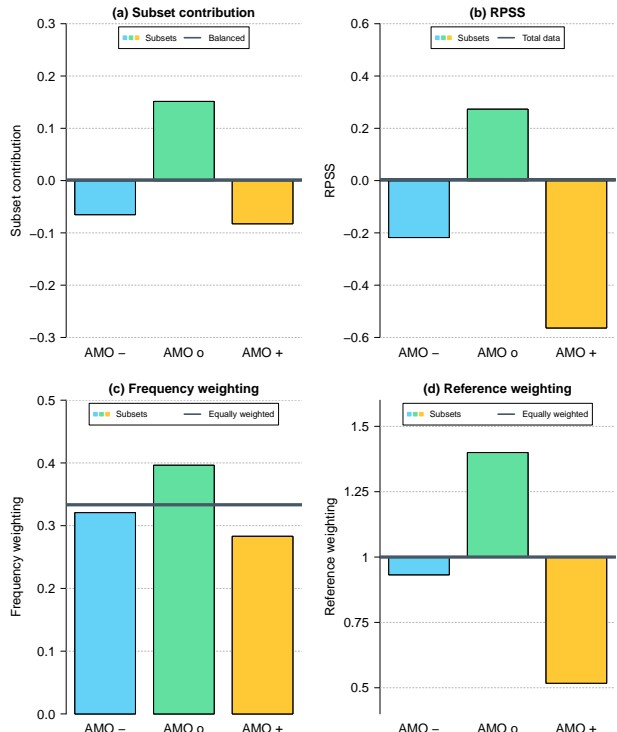

**Figure 7.** Same as Fig. 6, but with the *preop-LR* initialized prediction system as reference.

Since the reference weighting was not relevant in the above case, we now choose a reference system affected by the AMO phase: an earlier version of the decadal prediction system. The pre-operational version *preop* in a low-resolution configuration *LR* has a T63 horizontal grid ($1.875°$) and 47 vertical levels in the atmospheric component, while the ocean part has a horizontal resolution of $1.5°$ and 40 vertical levels. Being an older version, the low-resolution system is forced by CMIP5 external forcing (Giorgetta et al., 2013). The other settings (e.g., initialization and assimilation procedure etc.) remain unchanged compared to the *preop-dcpp-HR* version introduced in Sect. 4.1. Figure 7b shows again the RPSS of the individual subsets (bars) and the total RPSS as horizontal gray line. As the latter coincides with the zero skill score line, we see that the initialized prediction system *preop-dcpp-HR* does not outperform the low-resolution version *preop-LR* over the entire period. The subset RPSS under positive AMO conditions during the initialization procedure is strongly negative (-0.55); a similar tendency can be seen during the negative phase (-0.2). Only during the neutral AMO phase, the *preop-dcpp-HR* shows an improvement compared to the earlier system version.

Since classification of AMO phases is again based on ORA-S4, the frequency weighting terms are the same as in the previous case. Again, the weighting factor of the neutral AMO phase is slightly higher than that of the other two phases (Fig. 7c). The reference weighting exhibits huge differences among the individual phases (Fig. 7d). While the subset of the neutral AMO phase shows a weighting factor of 1.4, which is approximately 40 % higher than the balanced value (1, gray horizontal line),



the reference weighting term of the subset of the positive phase is 0.5 and thus only half of the balanced one. The reference weighting associated to the negative AMO phase (0.9) lies in between.

The individual subset contributions (Fig. 7a) are now affected by all three terms of the skill score decomposition. In particular, the reference weighting now influences the contribution to a large extent. While the subset RPSS (Fig. 7b) suggests a strong negative contribution to the overall skill driven by a positive AMO phase, the subset contribution (Fig. 7d) allows a slightly different interpretation: positive as well as negative AMO phases contribute negatively to the overall skill score with similar amounts, counteracting the benefits from the neutral AMO phase.

## 5 Summary and discussion

We present a decomposition of skill scores into contributions from subsets of the forecasts which are selected according to characteristics of processes or large scale circulation, climate states during initialization of the forecast system, seasons or regions. Using the MSESS and RPSS we give examples for this decomposition in the context of a synthetic data set designed to reveal situations where this decomposition shows its usefulness. To achieve this, the synthetic time series show different performance characteristics of forecast and reference systems in two subsets. These subsets contribute differently to the overall skill score in an additive way; according to their size, the performance of the forecast system on the subset and the performance of the reference system on the subset compared to the full data set. Hence, the contribution of a specific subset to the overall skill can be decomposed into

**subset skill score** $\mathrm{SS}_i = \frac{\mathrm{S}_i^{\mathrm{fc}} - \mathrm{S}_i^{\mathrm{ref}}}{\mathrm{S}^{\mathrm{perf}} - \mathrm{S}_i^{\mathrm{ref}}}$,

**frequency weighting** $W_{\mathrm{freq}_i} = \frac{N_i}{N}$,

**reference weighting** $W_{\mathrm{ref}_i} = \frac{\mathrm{S}^{\mathrm{perf}} - \mathrm{S}_i^{\mathrm{ref}}}{\mathrm{S}^{\mathrm{perf}} - \mathrm{S}^{\mathrm{ref}}}$.

The *subset skill score* measures the performance of a forecast system compared to a reference system for a particular subset, a useful and popular quantity to assess varying performance of a forecast system over different subsets; this is frequently used to detect enhanced/reduced predictability for certain climate and large-scale circulation states or specific seasons and regions (see references mentioned in the introduction). The *frequency weighting* reflects the size of the subset with respect to the full data set. For small subsets, it reduces the subset's contribution to the overall skill and vice versa for large subsets. The *reference weighting* adjusts the scale (or range) of the skill score, which is set by the difference between the reference performance of the subset and the perfect forecast, to the scale relevant for the overall data set. For negatively oriented scores with $S^{\mathrm{perf}} = 0$, this is expressed by the ratio of the two differences, see Eq. (3). *Reference weighting* as well as *frequency weighting* are both independent of the forecast system. The product of all three terms yields the subset's contributions to the overall skill score.

We expect that this decomposition helps to avoid mis-interpreting a potential performance increase in a subset resulting, e.g. from a significant performance decrease in the reference system. In this context, climatological forecasts used as a reference system could also impact the interpretation of the skill, as discussed for example in Hamill and Juras (2006).



Subsequently, we exemplify the RPSS decomposition in the context of the MiKlip decadal prediction system stratified along characteristics of the AMO during forecast system initialization. Target is the quantification of hindcast skill for the near-surface air temperature for leadyear 2–5 over the North Atlantic and European region. The initialized hindcasts (*preop-dcpp-HR*) show a weak positive overall skill (locally significant) in the north-eastern part of the North Atlantic and in the eastern Mediterranean compared to un-initialized hindcasts. Stratified verification along positive, negative and neutral AMO phases for initialization reveals

– a negative subset contribution to the total RPSS in the south-western North Atlantic and Scandinavia for a subset associated with neutral AMO and

– a positive subset contribution for W-EU and C-EU (AMO-) and in the North Atlantic (AMO+) for subsets associated with negative and positive AMO.

The decomposition for the Western European North Atlantic box shows that subsets associated with a positive AMO phase initialization contribute strongly to the positive total RPSS with a positive subset skill score only slightly modified by the frequency weighting.

Additionally, evaluation of the decadal hindcast system versus an earlier version with lower resolution (*preop-LR*) shows that individual subset contributions being affected by all three terms of the decomposition, with the reference weighting playing a particular role. This leads to a slightly different conclusion: While the subset RPSS suggests that the strong negative contribution to the overall skill is mainly driven by positive AMO initialization, the decomposition reveals that both, the negative as well as the positive AMO phases contribute negatively with the same amount, counteracting the benefits from the neutral AMO phase.

In summary, the recent hindcast outperforms the un-initialized simulations in the Western European North Atlantic sector mostly due to performance gain for positive AMO phase initialization. No overall performance benefits can be seen here with respect to *preop-LR*, but positive contributions are achieved when initializing during neutral AMO phases. The first findings are similar to those from Borchert et al. (2018) since AMO/AMV phases are linked to OHT with a lag of 5-10 years. Nevertheless, a stratification along different OHT states may strengthen the distinction between each subsets. As this paper focuses on suggesting the framework of skill score decomposition for stratified verification, analysis of physical processes being responsible for varying skill is beyond the scope of this study.

## 6 Conclusions

The verification of forecast systems stratified along characteristics of physical processes, large-scale circulation, climate states at initialization, seasons or regions can be a helpful tool for model development, the detailed assessment of forecasts quality, as well as for communication of forecasts. However, interpretation and comparison of skill scores across different strata can be challenging. This is not only the case for different subset sizes (*frequency weighting*) but also if the performance of the reference system varies strongly across subsets (*reference weighting*).



Both examples, the synthetic data set, as well as the one from decadal forecasting, exemplify the potential of the skill score decomposition for stratified verification. For the decadal prediction system initialized during positive AMO phases, we see a degradation of performance in the associated subset compared to its predecessor system. However, since the error of the reference system compared to observation in that subset is quiet small compared to the entire time series anyway (as can be seen from the reference weighting), the negative AMO phase negatively affects the overall performance in the same way as the positive AMO phase. In practice, potential model diagnostics and improvements should focus on both phases, rather than examining only the positive AMO phase suggested by the subset skill score assessment.

The skill score decomposition into contributions from suitable chosen subsets helps understanding possible model mis-behavior in a detailed and robust way as subsets can be chosen along the characteristics of physical processes. This yields valuable information for refinement of the forecast system or model development. Besides the state of the ocean or other large-scale conditions, seasonal as well as regional or other aspects can be addressed. Conditional or stratified verification can be used to investigate known or hypothetical linkages in the area of climate and weather forecasts including the ability to simulate and represent specific feedback mechanism. The example above examines the potential source of long-term predictability forced by certain ocean states associated with the AMO.

Finally, to support decision-making related to weather and climate, operational forecasts can be optimized by assessing and communicating its credibility in a more specific and situative way using stratified evaluation along conditions of initialization and the related skill score decomposition. Depending on the condition during initialization, forecast uncertainty can be quanti-fied and eventually the forecast can be rated as more precise, as addressed in Borchert et al. (2019). Similar is the identification of windows of opportunity for enhanced skill on subseasonal to decadal time scales (Mariotti et al., 2020). In the example from decadal forecasting, a better temperature forecast ability of the prediction system compared to the un-initialized one is achieved over parts of the North Atlantic for initialization during positive AMO phases.

The skill score decomposition framework suggested and exemplified in context of conditional or stratified verification is a relatively simple tool to analyze physical processes related to certain subsets and consequently supports the model development as well as the optimization of operational forecasts and their communication.

*Code and data availability.* The code used for the verification of decadal predictions is written in Shell and R and uses CDO. R is a GNU licensed free software from the R Project for Statistical Computing (http://www.r-project.org; last access: 11 January 2024). Cli-mate Data Operators (CDO, https://code.mpimet.mpg.de/projects/cdo; last access: 11 January 2024) is open source and released under the 3-clause BSD License. It is implemented as a software routine (*ProblEMS* plug-in) in the Freva system (Kadow et al., 2021) at Deutsches Klimarechenzentrum (DKRZ) and is versioned in gitlab. The version 1.6.3 used in this study is publicly available at https://doi.org/10.5281/zenodo.10469658. Synthetic examples, simulation data used in the conditional verification, and computed AMO time series (including computational routines) are publicly available at https://doi.org/10.5281/zenodo.10471224. HadCRUT4 data is freely available at https://www.metoffice.gov.uk/hadobs/hadcrut4/data/current/gridded_fields/HadCRUT.4.6.0.0.median_netcdf.zip (last access: 11 January 2024) and ORA-S4 ocean reanalysis data at https://icdc.cen.uni-hamburg.de/thredds/aggregationOras4Catalog.html?dataset=oras4_temp_all (last access: 11 January 2024).





*Author contributions.* AR prepared the manuscript with contributions from HR and JG. AR computed analyses and produced figures. HR, JG and AR developed the concept of the methodology. HR acquired the funding.

*Competing interests.* The authors declare that they have no conflict of interest.

*Acknowledgements.* We acknowledge funding from the Federal Ministry of Education and Research in Germany (BMBF) through the research project MiKlip II (grant numer (FKZ): 01LP1520A). This study used resources of Deutsches Klimarechenzentrum (DKRZ) and accessed data and plug-ins trough the Central Evaluation System framework Freva.




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
