# Peer review of "Decomposition of skill scores for conditional verification – Impact of AMO phases on the predictability of decadal temperature forecasts"

_EGUsphere, 2023_

## Referee Comment (RC2)

Review for "Decomposition of skill scores for conditional verification -Impact of AMO phases on the predictability of decadal temperature forecasts" by Richling et al

Richling et al propose a decomposition of mean skill scores as weighted sums $SS = \sum_i^{\mathcal{D}} W_i SS_i$ of the skill scores $SS_i$ for non-overlapping subsets $\{i \subset \mathcal{D} : \uplus i = \mathcal{D}\}$ of the data, with the weights $W_i$ given by the proportion of the data in each subset times the performance of the reference forecast for each subset relative to that for the full data. The decomposition is straightforward, as it derives from the associative property of addition. The authors use toy examples to examine how the weights $W_i$ modulate the skill score contributions $SS_i$ to the overall skill score over $\mathcal{D}$, and implement this methodology on predictions of 2m air temperature with the MiKlip system conditional to the 3 phases of the Atlantic Multidecadal Oscillation (AMO).

I agree that such an approach could be helpful to provide insights when evaluating forecast mean skill scores, and, despite its simplicity, I'm not aware of such a decomposition discussed elsewhere. However, the paper needs substantial improvement and can be made more concise. I thus recommend the authors to address the following comments before their paper can be considered for publication in GMD.

**Major**

1. The key to the decomposition in Eq. 2 is that, for the verification score $\mathrm{S}_n = \mathrm{S}(f_n, o_n)$ and the mean score in Eq. 1 (denoted $\overline{S}$ here), we have:

$$\overline{\mathrm{S}} = \frac{1}{N} \sum_{n=1}^{N} \mathrm{S}_n = \sum_{i=1}^{K} \frac{N_i}{N} \left( \frac{1}{N_i} \sum_{n=1}^{N_i} \mathrm{S}_n \right) = \sum_{i=1}^{K} \frac{N_i}{N} \overline{\mathrm{S}}_i \tag{1}$$

with $N = N_1 + \ldots + N_K$. I suggest indicating this in Eq. 1 to guide/justify the abrupt second equality in Eq. 2. Note the use of the overline to indicate the mean over sample. In the paper, both verification score and mean score are denoted the same, which is confusing.

2. Section 2 describing the decomposition of skill scores is too long. It is divided into 6 subsections, which I don't think is necessary. Consider a smaller section showing Eq. 1-3 and giving short descriptions for each term in the decomposition (no need to repeat the expressions that define each term in separate subsections). For example, L118-121 in subsection 2.2.4 can be moved right after Eq. (3), indicating which term in Eq. 3 is referring to, and each term then named and described. Also, statements like (L102-103) "In Sect. 3, this term can be found ...", or (L107-109) "Consequently, a situation..." can be deleted as they do not seem to add much to the description. The bottom line is that the decomposition is straightforward and each term is self-evident, so they do not need much discussion.

3. Section 3 describing the toy example can be shortened too. In particular, Figs. 1, 2 and 4 can be condensed into one easy-to-read 4-panel figure for cases A0-A2 and B0-B2, indicating

on the figure the terms in the decomposition (Eq. 3). For example, panel 1 and 2 can show the two panels in Fig. 1 for $SS_1$, $SS_2$ and $SS$, whereas panel 3 can show the panel in Fig. 2 for $W_{ref_1}$ and $W_{ref_2}$. Because $W_{freq_i} = 0.5$ for $i = 1, 2$, there is no need to show these, but can be mentioned in the caption too. Finally, panel 4 can show Fig. 4 for the contributions $W_1 SS_1$ and $W_2 SS_2$. Figure 3 doesn't seem to add much insight and could be deleted.

4. It is unclear whether any of the contributions to forecast skill from the 3 subsets is statistically significant. Given the short period available, each subset has about 17 years on average (the actual number of years is different for each subset and determines the frequency weighting). These are small samples and it is unclear whether the results and conclusions are robust. Can the authors comment on this? I suggest adding confidence intervals to the barplots of Figs. 1, 2 and 4 (perhaps condensed into one figure; see comment # 3), and stippling for the maps of Fig. 5b-d. Perhaps this could be done with the bootstrapping method used for Fig. 5a.

5. L249-250 The authors removed the linear trend from the SST average over the North Atlantic region to define the AMO index. It is known that this approach confound the internal variability of AMO with the underlying forcing signal from greenhouse gases and aerosols. A common and easier approach to derive the unforced index is to remove the SST global mean anomaly from that in the AMO region (e.g., Trenberth et al 2006 [`doi.org/10.1029/2006GL026894`]). Or, perhaps an even more accurate approach is that of Deser and Phillips (2021) [`doi.org/10.1029/2021GL095023`]. Do the results/conclusions for the MiKlip decadal system assessed here depend on how the AMO index is computed?

6. The three subsets in the analysis are determined by the phase of the AMO index in the observation-based dataset. However, it would be useful to know how the AMO is represented in the MiKlip system itself. I suggest to include an evaluation of the AMO in the decadal predictions, or provide a reference if this was done elsewhere. The point is, how relevant is the analysis if the forecasting system is unable to represent the AMO?

**Minor**

1. Please use a continuous line numbering to facilitate future revisions.

2. L4 Aim → The aim

3. L5-7 Consider rephrasing. To be precise, the overall skill score is decomposed into a sum, where each term of the sum is the product of 3 components as described. It is more telling perhaps to say that the overall skill score is decomposed into a weighted sum, where each term is ... (and then can go on about describing the weights and partial skill scores).

4. L10 Atlantic Meridional Oscillation → Atlantic Multi-decadal Oscillation

5. L12 due to performance gain → due to contributions

6. L13 a positive AMO phase → the positive AMO phase

7. L14-16 Delete "sophisticated". Perhaps "insightful" instead?

8. L25 ccore → score

9. Section 3 is described as showing results for "synthetic time series". Unless I missed it, there are no such time series, and the example simply feeds values conveniently into the decomposition of Eq. (3), as per Tables 1 and 2. If that is the case, please clearly indicate so and avoid the somewhat misleading terminology "synthetic time series".

10. I may have missed it, but clearly specify early on in Section 3 that for the toy example $S^{\text{perf}} = 0$.

11. L84 Delete "etc"

12. L87 is → are

13. L112 ajust → adjust

14. L123 degeneration → degradation

15. L128 What synthetic dataset? See item 9 above.

16. L140 Delete "assumption"

17. L150-151 Rephrase or delete "As a first guess from seeing the skill scores ..." Why one would think so? It is trivial that the sum of the skill scores in the subsets is not the skill score over the full set, nor the arithmetic mean in general. What is somewhat less clear is what the weighting is, which is addressed in the paper.

18. L159 we simulate → we consider (?)

19. L193 subset $i = 2$ ($SS_1$) → subset $i = 2$ ($SS_2$)

20. middle; → middle.

21. L200-201 Isn't the increase from -0.36 to 0.18, with $\Delta SS_1 = 0.54$ instead?

22. L220 componentes → components

23. L241 Typically, "hindcasts" refer to retrospective forecasts, which are model runs initialized from observation-based climate states, whereas "uninitialized predictions" refer to historical simulations (for past climate) or projections (for future climate) which are not initialized from observation-based climate states and have internal variability that are not expected to match observations. I recommend using this terminology and change "both hindcast sets" to, e.g., "both sets of predictions". This applies to other cases throughout the text. In particular, "initialized decadal hindcast" is redundant and "uninitialized hindcast" should be avoided (L284).

24. L251 time period → period (this applies to other instances in the text)

25. L259 Can the authors be more specific on how they divide the datasets? In particular, is the ensemble mean used to determine the terciles for hindcast and simulations?

26. L265 How is $Y_{j,t}$ obtained. Is it by simply counting the ensemble members in each category (then dividing by the ensemble size) for a given initial year? Please clarify.

27. L291 for Fig. 5b-d, please clarify if these "contributions" refer to $W_i SS_i$ or just $SS_i$. Clarify also in the caption to Fig. 5.

28. L293 W-EU and C-EU haven't been defined. They are defined in L296.

29. L297 "... with certain AMO phases identified in previous studies". Provide references.

30. L302 "a ... RPSS of 0.3 is achieved" → "RPSS=0.3". At the very least, delete "clearly positive".

31. L307 Given that there is contention on whether AMO may be influenced/determined by external forcing (e.g., Mann et al 2020 [`https://doi.org/10.1038/s41467-019-13823-w`]), and because the AMO phases used here are from the observation-based data, perhaps rephrase to "... uninitialized reference is not influenced by the AMO phases in the observations".

32. L309 Fig. 6a → Fig. 6b (?)

33. L310 Fig. 6d → Fig. 6c (?)

34. L311 Is this contribution statistically significant? A value of 0.08 doesn't seem like a "large amount", even though it is larger than the other two cases. See Major comment #4.

35. L359 Target → The goal (?)

36. L368 Here and elsewhere the authors use terminology like "positive AMO phase initialization". This is unclear. Consider changing to e.g., "forecast initialization during the positive phase of AMO".

37. L380 I may have missed it, but I don't think OHT was defined before.

38. L393 Delete "quite" and "anyway"

39. L391-396 Rephrase. This statement is convoluted and can be made clearer.

40. L406-407 Can the authors expand on how this work relates to: "forecast uncertainty can be quantified and eventually the forecast can be rated as more precised"? I fail to see a clear connection between this work and the quantification of forecast uncertainty.

---

## Author Comment (AC1)

We thank the two reviewers for reading our article carefully and providing constructive feedback and apologize for the late reply. We have revised the manuscript to account for their suggestions and insightful thoughts. The useful feedback helped to improve the manuscript's quality. As a major change, we re-designed and condensed the section describing the synthetic cases and adapted the structure of the skill score decomposition. The detailed responses are provided in the attached file. We provide a one-to-one response to all points raised by the reviewer. The reviewers' comments appear in black font and our responses in blue. All line numbers in the response documents refer to the lines in the revised manuscript without track changes.

**Additional changes:**

In addition to the changes according to the referee comments, we changed the following:

- We removed the references that were mistakenly listed in the old manuscript but not mentioned in the text:

  - Jungclaus et al. (2019): MPI-M MPI-ESM1.2-HR model output prepared for CMIP6 CMIP historical, https://doi.org/10.22033/ESGF/CMIP6.6594

  - Pohlmann et al. (2019): MPI-M MPIESM1.2-HR model output prepared for CMIP6 DCPP, https://doi.org/10.22033/ESGF/CMIP6.768

- We updated the data and software reference (incl. DOI) according to the changes made in the revised manuscript (l.427).

- We updated the websites for the Central Evaluation System mentioned in the manuscript (l.270ff).

- We did some minor corrections (typos, clarification).

**Reviewer comments:**

**RC1 comments:**

Richling, Grieger and Rust present a framework for the decomposition of skill scores stratified in subsets. The authors introduce the terms reference and frequency weighting to characterize the contribution of the subset skill scores to the grand total. The manuscript is well written and the application to decadal forecasting is illustrative. The decomposition is a useful addition to the existing forecast verification literature, however, the authors should expand the discussion of the method. In particular it is not clear, if the variability of the contributions is merely a consequence of the geometry of the problem (i.e. there being more room to improve when the reference forecast performs badly) or if we can learn something in excess of this (see also comments below). If the authors are willing to address this minor issue, I fully support publication of the manuscript.

We thank Jonas Bhend for the insightful comments and the useful suggestions. See below our response in detail.

General comments:

I: While I appreciate the synthetic test cases as a motivating example to elaborate the specifics of the decomposition, I think the setup could be improved for better interpretability. The authors already provide some motivation by mentioning 'what happens if we improve the forecast in subset 1'. Designing the synthetic cases more prominently along those lines would ease the interpretation. The synthetic setup could be altered to A/B0: base case, A/B1: improve SS1, A/B2: improve SS2.

Thanks for your advice to present the synthetic cases in a more prominent and easy-to-understand way. We followed your suggestion and replaced the cases to the

1) base case (A0/B0) and

2) a scenario with an improved skill score in subset 1 and

3) a scenario with an improved skill score in subset 2.

Changes are made in chapter 3.1 (l.123 – l.163)

II: Also, I strongly suggest to consider improvements of the score of equal (relative) size for better comparison of the effect. In particular I would be interested to see if the increased reference weighting is basically a consequence of there being more room to improve when the reference performs poorly. As such a synthetic set of experiments with differing reference weighting, but the same relative improvement in the respective subsets could be illustrative.

In the new designed synthetic cases (see comment RC1 I) we have considered your reasonable suggestion to improve the respective subsets with the same relative improvement, for a better comparison. In detail we improve the score of the forecast system in the subset by reducing the score to the half in comparison to the score in the base cases (A0/B0). Generally, this relative behavior is similar to that we have shown in the first version of the manuscript, because it also affects the subset skill scores in a similar way. A relative improvement of a subset with a substantial higher reference weighting affects the total skill score much more than the same relative improvement in the subset with a lower reference weighting (see Fig. 1 / Table 1/2). As a consequence, we agree that subsets with an increased reference weighting allow more potential for improvements when the reference performs poorly in that subset (see also RC1 2.).

III: The summary in the summary and discussion part is redundant. I suggest to remove or at least considerably shorten this as it doesn't add to the paper.

Thank you for the advice. We have removed the redundant summary paragraph and moved the important passage to the related results within the text (l.363ff). In addition, we have extended the discussion part (l.373ff, see RC2 IV/V)

IV: I encourage the authors to think of potential applications outside the domain of decadal forecasting to increase the appeal for readers outside of this community.

Thank you for the reasonable advice. We have added statements about potential applications in l.412ff

"A potential application outside the domain of decadal prediction could be the identification and analysis of such a window. In weather forecasting, the conditional verification stratified along particular flow regime conditions (e.g., blocking) or along different states of MJO and ENSO in subseasonal-to-seasonal predictions could be reasonable."

and l.307ff

"Outside of the field of decadal predictions, the simultaneous investigation of the terms could be useful to evaluate and interpret regionally (e.g., mountains and low-lands) or seasonally varying error behaviors with respect to the total model performance. A possible application is shown in Peter et al. (2024) using the example of the evaluation of statistical models for extreme precipitation."

Minor comments:

1. L3: Providing some more context with an illustrative example at the start of the abstract would improve readability.

Thank you for the advice. We added the following sentence at the beginning of the abstract (l.1ff):

"Since the performances of weather and climate forecasting systems and their competing reference systems are generally not homogeneous in time and space and may vary in specific situations, improvements in certain situations or subsets have different effects on overall skill."

2. L180ff: This implies that we benefit more from improvements in subsets for which the reference performs badly. With a mildly skillful reference, the reference score basically measures inherent predictability. Consequently, the above translates to we profit more from improvements in situations with limited predictability. If this can be supported, this would imply that we should focus more on subsets that are hard to predict if we want to improve skill in general. This seems contrary to what is usually being done, i.e. exploit situations with relatively high predictability (and plausible hypotheses on drivers) and try to improve predictions there. The authors mention in the conclusion that the focus should not only be with AMO+ situations. Maybe the predictability angle could provide some more grounds for the discussion of the implications of the decomposition.

Thank you for these insightful thoughts. If we assume that the reference score measures inherent predictability, then in terms of the overall performance, we would benefit more from improvements in subsets/situations with a higher reference weighting (limited predictability) because there is more potential room for improvement. On the other hand, it can be more difficult to improve the skill (of equal relative size) in these situations as the processes and drivers increasing the predictability may not be present or may have less of an effect. However, If there is the aim and opportunity to find new potential processes, drivers or links, then one could focus more on these subsets and try to improve the predictions there if we want to improve the skill in general. In the end, one has to balance the two cases for a decision, and our decomposition could be a helpful tool to support the assessment.

To address this, we implemented the following in l.392ff:

"Assuming the reference score could measure inherent predictability with a mildly skillful

reference, we would benefit more from improvements in subsets/situations with limited predictability (higher reference weighting) in terms of the overall skill. In contrast, improvements in situation with higher predictability have less effect on the total skill. However, it can be more difficult to improve the skill (of equal relative size) in these situations as the processes and drivers increasing the predictability may not be present or have less impacts. Accordingly, the decomposition can help to balance the aspects in order to support the assessment for a decision."

3. Figure 3: This is mildly confusing because different labels are used compared with the tables. To improve readability, the corresponding points could be labelled with A0, A1, A2, … and the labels could be replaced with Table A,B instead of Cases A,B (vertical lines) and Case A0/B0, A1/B1, A2/B2 instead of Case 1,2,3 (facets) for clarity.

Thank you for the advice. To improve the readability, we have implemented your suggestions. Instead of labeling the case examples of A and B to Table A and B, we have renamed it in the manuscript and the figures to setup A and setup B.

4. Figure 4: This figure feels somewhat redundant, maybe the contribution could be integrated with Tables A/B or Figure 1 for clarity.

We would like to keep the Figure showing the subset contributions. Later in the decadal prediction analyses, the term is used for the interpretation of the results. However, as you suggested, we have integrated the plot with Figure 1 and 2 for a better comparison and clarity (see also RC2 III).

5. L257: Are scores indeed computed on 4 yearly average temperatures (period 2-5 years), or are scores computed on monthly mean temperatures as specified in L242 and aggregated for the lead times 2-5 years?

The scores are computed on 4 yearly average temperatures (period 2-5 years). At the end of Sec. 4.1, we wanted to mention the data we get from HadCRUT4 is on a monthly basis. Before the scores are calculated, we build 4 yearly average temperature time series for lead-years 2-5. To avoid confusion, we removed "on the basis of monthly mean temperatures" at the end of Sec. 4.1. To make it clearer, we also replaced the sentence to "Temperature data with lead-times between 2 and 5 years are averaged to compute a score for the lead-time period 2--5 years." (l.246).

Editorial comments:

6: L4: The aim is to …

Corrected.

7. L21: is the comparison against another competing prediction system or a standard reference forecast such as the persistence or climatological forecast.

Suggestion implemented.

8. L24: … and the continuous ranked probability skill score (CRPSS) for probabilistic forecasts are widely used decadal forecast verification (e.g., Kadow et al., 2016; Kruschke et al., 2016; Pasternack et al., 2018, 2021).

Thank you for the suggestion. However, we would leave the sentence as it is, as the scores are also used outside the domain of decadal predictions. The whole paragraph is rather addressed to the general reader, with an exemplary mention of decadal predictions. Additionally, not all listed scores are covered in the mentioned studies on decadal predictions as the suggestion could imply.

9. L112: adjusts

Corrected.

10. L139: the mean scores of the forecast systems differ

Corrected.

11. L140: mean scores of the reference system

Corrected.

12. L179: in the same way (or in a similar way)

Corrected.

L189: Generally

Corrected.

13. L241: against monthly mean temperatures from the HadCRUT4 observation data set (Morice et al., 2012). [Also I suggest to refer to the obs data as HadCRUT4 consistently throughout (e.g. L255).]

To avoid confusion, we removed "on the basis of monthly mean temperatures". (see RC1 5.). As you suggested, we referred the observation data to HadCRUT4 consistently.

14. L251: annual averages

Corrected.

15. L289: with significant values patches with positive but non-significant skill are visible …

Corrected.

16. L307: uninitialized reference is not influenced …

Corrected.

17. L393: quite small

We rephrased according to RC2 39.

**RC2 comments:**

Richling et al propose a decomposition of mean skill scores as weighted sums $\mathrm{SS} = \sum_i^{\mathcal{D}} W_i \, SS_i$ of the skill scores $SS_i$ for non-overlapping subsets $\{i \subset \mathcal{D} : \uplus i = \mathcal{D}\}$ of the data, with the weights $W_i$ given by the proportion of the data in each subset times the performance of the reference forecast for each subset relative to that for the full data. The decomposition is straightforward, as it derives from the associative property of addition. The authors use toy examples to examine how the weights $W_i$ modulate the skill score contributions $SS_i$ to the overall skill score over $\mathcal{D}$, and implement this methodology on predictions of 2m air temperature with the MiKlip system conditional to the 3 phases of the Atlantic Multidecadal Oscillation (AMO).

I agree that such an approach could be helpful to provide insights when evaluating forecast mean skill scores, and, despite its simplicity, I'm not aware of such a decomposition discussed elsewhere. However, the paper needs substantial improvement and can be made more concise. I thus recommend the authors to address the following comments before their paper can be considered for publication in GMD.

We thank the reviewer for the careful reading, comments and useful suggestions. See below our response in detail.

Major

I: The key to the decomposition in Eq. 2 is that, for the verification score $S_n = S(f_n, o_n)$ and the mean score in Eq. 1 (denoted $\overline{S}$ here), we have:

$$\overline{S} = \frac{1}{N} \sum_{n=1}^{N} S_n = \sum_{i=1}^{K} \frac{N_i}{N} \left( \frac{1}{N_i} \sum_{n=1}^{N_i} S_n \right) = \sum_{i=1}^{K} \frac{N_i}{N} \overline{S}_i \qquad (1)$$

with $N = N_1 + \ldots + N_K$. I suggest indicating this in Eq. 1 to guide/justify the abrupt second equality in Eq. 2. Note the use of the overline to indicate the mean over sample. In the paper, both verification score and mean score are denoted the same, which is confusing.

We thank you for the hint and the advice for better guidance with equations for the decomposition. We implemented the suggested equation into Eq. 1 and indicated the mean score with an overline in

the whole paper to avoid confusion. We also implemented "non-overlapping" for a clearer definition of the subsets in l.79.

II: Section 2 describing the decomposition of skill scores is too long. It is divided into 6 subsections, which I don't think is necessary. Consider a smaller section showing Eq. 1-3 and giving short descriptions for each term in the decomposition (no need to repeat the expressions that define each term in separate subsections). For example, L118-121 in subsection 2.2.4 can be moved right after Eq. (3), indicating which term in Eq. 3 is referring to, and each term then named and described. Also, statements like (L102-103) "In Sect. 3, this term can be found ...", or (L107-109) "Consequently, a situation..." can be deleted as they do not seem to add much to the description. The bottom line is that the decomposition is straightforward and each term is self-evident, so they do not need much discussion.

Thank you for the advice. Generally, we have shortened and condensed this section (Sect. 2) according to your suggestion. However, we would like to keep a more detailed description of each term for readers who are not as familiar with the mathematical construction and interpretation of skill scores.

III: Section 3 describing the toy example can be shortened too. In particular, Figs. 1, 2 and 4 can be condensed into one easy-to-read 4-panel figure for cases A0-A2 and B0-B2, indicating on the figure the terms in the decomposition (Eq. 3). For example, panel 1 and 2 can show the two panels in Fig. 1 for $SS_1$, $SS_2$ and $SS$, whereas panel 3 can show the panel in Fig. 2 for $W_{ref_1}$ and $W_{ref_2}$ . Because $W_{freq_i} = 0.5$ for $i = 1$, 2, there is no need to show these, but can be mentioned in the caption too. Finally, panel 4 can show Fig. 4 for the contributions $W_1 SS_1$ and $W_2 SS_2$. Figure 3 doesn't seem to add much insight and could be deleted.

Thank you for the suggestion. We condensed Fig. 1, 2 and 4 (old manuscript version) to one 4-panel figure (Fig. 1 revised manuscript) for a better comparison and shortened the text. However, we think Figure 3 (old manuscript version) well demonstrates, in an illustrative and condensed way, the potential influence of the different reference weighting on the total skill score applied to all shown example cases. Therefore we would like to keep the Figure.

IV: It is unclear whether any of the contributions to forecast skill from the 3 subsets is statistically significant. Given the short period available, each subset has about 17 years on average (the actual number of years is different for each subset and determines the frequency weighting). These are small samples and it is unclear whether the results and conclusions are robust. Can the authors comment on this? I suggest adding confidence intervals to the barplots of Figs. 1, 2 and 4 (perhaps condensed into one figure; see comment # 3), and stippling for the maps of Fig. 5b-d. Perhaps this could be done with the bootstrapping method used for Fig. 5a.

Indeed, the sample of each subset has a size of 17 years on average. Since the focus of this study is to demonstrate the methodology of the decomposition into the different terms and its interpretation in general, we have not considered the aspect of uncertainties here. With respect to comment RC2 V, we agree that a more robust analysis should consider more factors, including also statements about the uncertainties. In a recent project, it is planned to focus more in detail on the uncertainty

for each term and consider various definitions of the ocean state to be more robust. To be transparent, we also added a statement in the Discussion Sect. (l.373ff):

"Since our study does not fully account for uncertainties and the results are partly sensitive to the defined W-EU NA region and the chosen AMO index representing the ocean state (see supplementary material), further indices and sensitivity studies including the consideration of uncertainties can be applied for a more robust analysis."

However, since the synthetic example cases are given by mean values, we can define the values in a way that uncertainties can be neglected. So there is no need for confidence intervals in Fig1, 2 and 4 (old manuscript; condensed into Fig. 1 (a-d) in the revised version). For the subset contributions in Fig. 3b-d (revised version) we have highlighted significant values based on the bootstrap method used for Fig. 3a (revised version) as you suggested. Additionally, in Fig 4/5a (revised version) we show 95 %-confidence intervals for the subset contributions. In addition, we mention the significance in the corresponding parts of the text.

V: L249-250 The authors removed the linear trend from the SST average over the North Atlantic region to define the AMO index. It is known that this approach confound the internal variability of AMO with the underlying forcing signal from greenhouse gases and aerosols. A common and easier approach to derive the unforced index is to remove the SST global mean anomaly from that in the AMO region (e.g., Trenberth et al 2006 [doi.org/10.1029/2006GL026894]). Or, perhaps an even more accurate approach is that of Deser and Phillips (2021) [doi.org/10.1029/2021GL095023]. Do the results/conclusions for the MiKlip decadal system assessed here depend on how the AMO index is computed?

We agree the used AMO index could be defined in other ways to avoid confounding the internal variability of AMO with the underlying forcing signal. In our analyses, we also applied the AMO index suggested from Trenberth et al. (2006). The overall tendencies of the subset contributions shown in Fig. 3 in the revised manuscript (also Fig. R1a in this document) look similar for this AMO index (Fig. R1b). For the Western European North Atlantic box (Fig. 4/5 in revised manuscript, Fig. R2/3a) we have computed the skill score and decomposition terms using the Trenberth AMO index, but for a slightly extended region to the west (45–10° W, 35–60° N) to also include regional sensitivity. There (Fig. R2/3b) we get partly different results and conclusions.
It shows that our results are sensitive to the defined AMO index as well as the specific region box we analyzed. We agree that a more robust analysis should consider more factors. The focus of the example from the field of decadal prediction is still to demonstrate the possible application of the stratified verification. However, to address the aspect of the dependency of our results (in combination with the uncertainty aspect addressed in RC2 IV), we added the following text to the Discussion Section (l.373ff) and provided the additional figures in the supplementary material.

"Since our study does not fully account for uncertainties and the results are partly sensitive to the defined W-EU NA region and the chosen AMO index representing the ocean state (see supplementary material), further indices and sensitivity studies including the consideration of uncertainties can be applied for a more robust analysis."

**AMO (Enfield et al., 2001)**

[Figure]

**AMO (Trenberth et al., 2006)**

[Figure]

**Figure R1a)** Figure as Fig. 3 in the revised manuscript, but without showing significant areas for subset contributions (b-d).

**Figure R1b)** Same as Fig. R1a, but using the AMO index according to Trenberth et al. (2006) to define the subsets.

[Figure]

[Figure]

**Figure R2a)** Figure as Fig. 4 in the revised manuscript, but without showing confidence intervals for subset contributions (a).

**Figure R2b)** Same as Fig. R2a, but using the AMO index according to Trenberth et al. (2006) and a Western European North Atlantic (W-EU NA) box slightly extended to the west (45–10° W, 35–60° N).

[Figure]

[Figure]

**Figure R3a)** Figure as Fig. 5 in the revised manuscript, but without confidence intervals for subset contributions (a).

**Figure R3b)** Same as Fig. R3a, but using the AMO index according to Trenberth et al. (2006) and a Western European North Atlantic (W-EU NA) box slightly extended to the west (45–10° W, 35–60° N).

VI: The three subsets in the analysis are determined by the phase of the AMO index in the observation-based dataset. However, it would be useful to know how the AMO is represented in the MiKlip system itself. I suggest to include an evaluation of the AMO in the decadal predictions, or provide a reference if this was done elsewhere. The point is, how relevant is the analysis if the forecasting system is unable to represent the AMO?

Thanks for mentioning this reasonable point. Some studies have already shown that the multi-decadal state of the ocean in the North Atlantic (e.g., AMV, AMOC, OHT) is represented in the decadal prediction system. Since the AMO, also known as AMV, is linked to AMOV and OHT as mentioned in the introduction (Müller et al., 2014; Zhang and Zhang, 2015; Borchert et al., 2018, 2019), we did not explicitly evaluate the AMO in this system. However, we think it is reasonable to add this aspect to the text. We rephrased (l.232ff) to "Since the multi-decadal variability of the ocean state in the North Atlantic (e.g., AMV, AMOC, OHT) is represented in the decadal prediction system and shows predictive potential (Müller et al., 2014; Borchert et al., 2018, 2019; Höschel et al., 2019), we will apply the conditional verification of the temperature stratified along three different phases of the Atlantic Multidecadal Oscillation (AMO). ..."

Minor

1. Please use a continuous line numbering to facilitate future revisions.

Thank you for the hint. We used the official tex template provided by Copernicus. In the revised manuscript we used an updated version of the template.

2. L4 Aim → The aim

Corrected.

3. L5-7 Consider rephrasing. To be precise, the overall skill score is decomposed into a sum, where each term of the sum is the product of 3 components as described. It is more telling perhaps to say that the overall skill score is decomposed into a weighted sum, where each term is ... (and then can go on about describing the weights and partial skill scores).

Thank you for the careful advice. We rephrased to (l.5ff):

"The overall skill score is decomposed into a weighted sum representing subset contributions, where each individual contribution is the product of: (1) the subset skill score assessing ..."

4. L10 Atlantic Meridional Oscillation → Atlantic Multi-decadal Oscillation

Thank you! Corrected.

5. L12 due to performance gain → due to contributions

The decadal prediction system performs better than the historical simulations in terms of the overall skill score. Since the subset contribution of the positive AMO is driven mostly due to the subset skill score, we want to avoid the term "contributions" and want to be more specific. We have rephrased to (l.12ff): "… mostly due to contributions during the positive AMO phase driven by the subset skill score."

6. L13 a positive AMO phase → the positive AMO phase

Corrected.

7. L14-16 Delete "sophisticated". Perhaps "insightful" instead?

Thanks, we implemented your suggestion.

8. L25 ccore → score

Corrected.

9. Section 3 is described as showing results for "synthetic time series". Unless I missed it, there are no such time series, and the example simply feeds values conveniently into the decomposition of Eq. (3), as per Tables 1 and 2. If that is the case, please clearly indicate so and avoid the somewhat misleading terminology "synthetic time series".

Thanks for the hint. At the beginning of Sec. 3.1 (l.124ff), we assume a data structure like a time series to define the mean scores and skill score cases to fit into the narrative of time-based stratified verification. To avoid confusion, we changed the terminology to "Synthetic cases".

10. I may have missed it, but clearly specify early on in Section 3 that for the toy example $S^{perf} = 0$.

It was mentioned later in section 3.2. We now specified it at the beginning of Sect. 3 l.137).

11. L84 Delete "etc"

Deleted.

12. L87 is → are

Corrected.

13. L112 ajust → adjust

Corrected.

14. L123 degeneration → degradation

Corrected.

15. L128 What synthetic dataset? See item 9 above.

Thanks for mentioning this issue here again. As you can see above (item 9), we rephrased the section to "synthetic cases".

Indeed, we have not prepared a synthetic data set which has the full structure of a forecast/hindcast, reference forecast and observations to finally compute the verification score and skill score values. We prepared synthetic data which we used to define the mean verification score for the different subsets. This led to skill score of the subsets, to the mean score and skill score for the whole period.

To avoid misunderstanding, we changed "synthetic data set" to "synthetic data"

16. L140 Delete "assumption"

Since we have changed the design of the synthetic cases (see RC1 I) we rephrased the text accordingly and the sentence no longer exists in this form.

17. L150-151 Rephrase or delete "As a first guess from seeing the skill scores ..." Why one would think so? It is trivial that the sum of the skill scores in the subsets is not the skill score over the full set, nor the arithmetic mean in general. What is somewhat less clear is what the weighting is, which is addressed in the paper.

We agree that the case is trivial for the community inside the field of statistics and verification. However, motivated by Simpson's Paradox, we think it may not be so clear at first impression to

the general public and other readers outside the field who are not as familiar with the mathematical construction of skill scores. For this reason we would keep the general formulation. However, we have rephrased to (l.141ff) "Following the Simpson's Paradox, from seeing the skill scores one might be tempted to think the total skill ...".

18. L159 we simulate → we consider (?)

Thanks for the suggestion. We changed "simulate" to "set".

19. L193 subset i = 2 ($SS_1$) → subset i = 2 ($SS_2$)

Corrected.

20. middle; → middle.

Corrected.

21. L200-201 Isn't the increase from -0.36 to 0.18, with $\Delta SS_1 = 0.54$ instead?

Yes, you are right. However, we have generally changed the design of the synthetic cases and replaced the values in the text accordingly.

22. L220 componentes → components

Corrected.

23. L241 Typically, "hindcasts" refer to retrospective forecasts, which are model runs initialized from observation-based climate states, whereas "uninitialized predictions" refer to historical simulations (for past climate) or projections (for future climate) which are not initialized from observation-based climate states and have internal variability that are not expected to match observations. I recommend using this terminology and change "both hindcast sets" to, e.g., "both sets of predictions". This applies to other cases throughout the text. In particular, "initialized decadal hindcast" is redundant and "uninitialized hindcast" should be avoided (L284).

Thank you for the advice. We implemented your suggestions and renamed to "initialized decadal simulations" and „un-initialized historical simulations". Generally, although it is redundant, we would like to keep the term "initialized" and "un-initialized" in some certain cases to make it easier for readers outside the field of decadal predictions to distinguish between the two simulations.

24. L251 time period → period (this applies to other instances in the text)

Corrected in the entire text.

25. L259 Can the authors be more specific on how they divide the datasets? In particular, is the ensemble mean used to determine the terciles for hindcast and simulations?

Thank you for the question. Instead of the ensemble mean, the entire ensemble is used to determine the terciles. We added the sentence (l.250): "For both simulation data sets, the entire ensemble is used to determine the respective terciles."

26. L265 How is $Y_{j,t}$ obtained. Is it by simply counting the ensemble members in each category (then dividing by the ensemble size) for a given initial year? Please clarify.

Thank you for the question. You are right, $Y_{j,t}$ is obtained by counting the ensemble members in each category and then dividing by the ensemble size for a given initial year. We added (l.257) „… by counting the ensemble members in each category and then dividing by the ensemble size" to the sentence for clarification.

27. L291 for Fig. 5b-d, please clarify if these "contributions" refer to $W_i SS_i$ or just $SS_i$. Clarify also in the caption to Fig. 5.

As described in Section 2.2 we use the terminology "subset contribution" for $W_i SS_i$ and "subset skill score" for $SS_i$ to distinguish between the two. For a better clarification we added $W_i SS_i$ and $SS_i$ in the text.

28. L293 W-EU and C-EU haven't been defined. They are defined in L296.

Thank you. We added the definitions.

29. L297 "... with certain AMO phases identified in previous studies". Provide references.

We added references (Zhang and Zhang, 2015; Borchert et al., 2018, 2019) and rephrased to "… with certain states of the ocean identified …" (l.290ff).

30. L302 "a ... RPSS of 0.3 is achieved" → "RPSS=0.3". At the very least, delete "clearly positive".

Thank you. We implemented the suggestion.

31. L307 Given that there is contention on whether AMO may be influenced/determined by external forcing (e.g., Mann et al 2020 [**https://doi.org/10.1038/s41467-019-13823-w**]), and because the AMO phases used here are from the observation-based data, perhaps rephrase to "... uninitialized reference is not influenced by the AMO phases in the observations".

Thank you for the hint. We have rephrased the sentence as you suggested.

32. L309 Fig. 6a → Fig. 6B (?)

Corrected.

33. L310 Fig. 6d → Fig. 6C (?)

Corrected.

34. L311 Is this contribution statistically significant? A value of 0.08 doesn't seem like a "large amount", even though it is larger than the other two cases. See Major comment #4.

As you mentioned, the "large amount" refers to the relative contribution to the total RPSS. To avoid confusion, we have rephrased the sentence (l.305ff) to: "The resulting ... positive AMO phase contributes the most (around 0.08) to the total RPSS, followed by the neutral AMO phase with a much smaller contribution of 0.02."

Regarding the significance, all three subset contributions are not statistically significant (95%-level), albeit very close for the subset of the positive AMO phase.

We added to the text (l.303f):

"Although the contributions show large uncertainties and are not statistically significant, tendencies can be derived." to the text at the beginning of the related paragraph.

35. L359 Target → The goal (?)

Corrected.

36. L368 Here and elsewhere the authors use terminology like "positive AMO phase initialization". This is unclear. Consider changing to e.g., "forecast initialization during the positive phase of AMO".

Thanks for the advice. We rephrased it here and elsewhere in the text to "… AMO phase at the time of the initialization".

37. L380 I may have missed it, but I don't think OHT was defined before.

It was already defined before (l.44).

38. L393 Delete "quite" and "anyway"

Deleted and rephrased.

39. L391-396 Rephrase. This statement is convoluted and can be made clearer.

Thank you for the comment, we have rephrased to (l.385ff):

"For the decadal prediction system, we see the strongest degradation of performance compared to its low-resolution system if it is initialized during positive AMO phases. However, the error of the reference system compared to observation in that subset is small compared to the entire time series (as can be seen in the lower reference weighting). As a consequence, the positive AMO phase negatively contributes to the overall performance nearly with the same amount as the negative AMO phase, although the subset skill score is much worse. In practice, potential model diagnostics and improvements should focus on both phases, rather than examining only the positive AMO phase suggested by the subset skill score assessment alone."

40. L406-407 Can the authors expand on how this work relates to: "forecast uncertainty can be quantified and eventually the forecast can be rated as more precised"? I fail to see a clear connection between this work and the quantification of forecast uncertainty.

Thank you for pointing this out. What we meant to say was with the stratified verification, one possible application could be to find subsets (e.g., periods) for which the forecast is more accurate and can be trusted more than for other periods. On the one hand, this could be reflected in the subset skill score. On the other hand, the uncertainties in this subset could also be smaller. Since we did not explicitly investigate the aspect of uncertainty in this study, we replaced the uncertainty aspect with "skill" in the sentence.